# Distribution of Gutless Siboglinid Worms (Annelida, Siboglinidae) in Russian Arctic Seas in Relation to Gas Potential

Nadezda P. Karaseva [1,*], Nadezhda N. Rimskaya-Korsakova [1], Roman V. Smirnov [2], Alexey A. Udalov [3], Vadim O. Mokievsky [3], Mikhail M. Gantsevich [1] and Vladimir V. Malakhov [1]

[1] Faculty of Biology, M.V. Lomonosov Moscow State University, 119991 Moscow, Russia
[2] Zoological Institute, Russian Academy of Sciences, 199034 St. Petersburg, Russia
[3] P.P. Shirshov Institute of Oceanology, Russian Academy of Sciences, 117997 Moscow, Russia
* Correspondence: oasisia@gmail.com

**Abstract:** In the Russian Arctic seas and adjacent areas of the Arctic basin, 120 sites of siboglinid records are currently known. Individuals belonging to 15 species have been collected. The largest number (49.2%) of records were made in the Barents Sea, followed by the Laptev Sea (37.5%) and the Arctic basin (10 records; 8.3%). No siboglinids have been reported from the Chukchi Sea. The largest number of species has been identified in both the Laptev Sea and Arctic basin (seven species each). Seventy-eight percent of the records were discovered at water depths down to 400 m. Many of the siboglinid records in the Arctic seas of Russia are associated with areas of high hydrocarbon concentrations. In the Barents Sea, *Nereilinum murmanicum* has been collected near the largest gas fields. The records of *Oligobrachia haakonmosbiensis*, *N. murmanicum*, *Siboglinum ekmani*, *Siboglinum hyperboreum*, *Siboglinum norvegicum*, as well as two undetermined species of siboglinids are associated with the marginal areas of bottom gas hydrates where methane emissions can occur. The Arctic seas of Russia feature vast areas of permafrost rocks containing gas hydrates flooded by the sea. Under the influence of river runoff, gas hydrates dissociate, and methane emissions occur. *Crispabrachia yenisey* and *Galathealinum karaense* were found in the Yenisei estuary, and *O. haakonmosbiensis* was found in the Lena estuary.

**Keywords:** methane seeps; gas hydrates; permafrost; Arctic warming

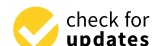



## 1. Introduction

The Siboglinidae is a family of sedentary annelid worms whose representatives all lack a digestive tract. The nourishment of siboglinids is provided by symbiotic bacteria. Within Siboglinidae, four groups of organisms are distinguished, differing in habitat and type of symbiotic bacteria [1]. Representatives of the genus *Osedax* Rouse, Goffredi & Vrijenhoek, 2004 contain heterotrophic symbionts that gain energy by splitting lipids contained in cetacean skeletons [2–4]. Vestimentiferans inhabit hydrothermal oases and areas of cold hydrocarbon seeps and have chemoautotrophic sulfide-oxidizing symbionts [1,5–7]. Metagenomic studies show that vestimentiferan symbionts are able to combine autotrophic and heterotrophic nutrition, i.e., to nourish themselves mixotrophically [8,9]. Small and thin Monilifera worms settle on sunken wood, ropes, and cardboard but can also occur in a silty substrate [10–15]. Data on the nature of moniliferan symbionts differ: some authors believe that they have methane-oxidizing symbionts [16,17], and others believe that they have sulfide-oxidizing symbionts [18,19].

Species of the Frenulata group (Pogonophora sensu stricto) inhabit soft sediments throughout the world's oceans in a wide range of depths—from sublittoral to hadal. Some species of Frenulata have been shown to harbor methane-oxidizing symbionts [20], while others harbor sulfide-oxidizing symbionts [21,22].

In areas of hydrocarbon seeps in the sediment column, the anaerobic oxidation of methane occurs using sulfates with the participation of free-living microorganisms [17,19,23–30]. This yields high concentrations of sulfide, which serves as an energy source for the sulfide-oxidizing symbionts of siboglinids.

The seas of the Russian Arctic sector are a region with large hydrocarbon resources [31–37]. This suggests a rich siboglinid fauna. This study provides an overview on the distribution of all known siboglinid records in the Arctic seas of Russia in connection with areas of hydrocarbon manifestations of various genesis. We expect that these data will allow us to understand the extent of our knowledge of the siboglinid fauna in the Arctic seas of Russia, which will serve as a basis for further faunal and hydrobiological studies. The obtained data may be useful for understanding to what degree the spreading of siboglinids is determined by the presence of methane seeps and gas hydrates.

## 2. Data Sources

Information on the distribution of siboglinids in the Russian sector of the Arctic was obtained from the analysis of literature sources [13,38–46]. We have also included data on new records of siboglinids in the St. Anna Trench in the Kara Sea and several new records of *Oligobrachia haakonmosbiensis* Smirnov, 2000 in the Laptev Sea, which have not yet been published [47–49]. These data were obtained from the collections of the Zoological Institute of the Russian Academy of Sciences and the P.P.Shirshov Institute of Oceanology of the Russian Academy of Sciences. Unpublished data on new records were obtained from collections during the Arctic expeditions of the NIS "Academician Mstislav Keldysh" in 2017–2021. In all expeditions, the material was collected in the same way, namely, with a Van Veen dredge with a capture area of 0.1 m$^2$. All selected material is fixed in 2.5% glutaraldehyde and 96% alcohol.

## 3. Overview of Siboglinid Records

1.      *Oligobrachia haakonmosbiensis* Smirnov, 2000

The type locality of *O. haakonmosbiensis* is the area of the Haakon Mosby mud volcano in the Norwegian Sea [13]. Within the region under consideration here, individuals of *O. haakonmosbiensis* are known in three seas: the Barents Sea, the Laptev Sea, and the East Siberian Sea, as well as in the Arctic Basin outside the official borders of the Barents Sea (Figure 1A,C,D). Within the described area, *O. haakonmosbiensis* occupies a very wide depth range (26–2166 m; Figure 2). Most of the records are concentrated in depths between 26 and 200 m (Figure 2).

In the Barents Sea, *O. haakonmosbiensis* is known from two records: one located south of the Svalbard archipelago on the border with the Norwegian Sea, and the other in the western trough on the border with the Medvezhinsko-Nadezhdinskaya Bank at depths of 380 and 350 m, respectively. Outside the Barents Sea, north of its official border, to the northeast of the Svalbard archipelago in the Arctic Basin, the deepest record for the species stems from 2166 m (Figure 1).

In the Laptev Sea, *O. haakonmosbiensis* is widely distributed in the shelf part at depths of 50–300 m, mostly from 50–100 m (Figure 1C). *O. haakonmosbiensis* is the only representative of siboglinids in the East Siberian Sea (Figure 1D). There, it was found at two stations (26 and 48 m depth). These are the shallowest locations within its known range. The records of *O. haakonmosbiensis* in the East Siberian Sea are the easternmost records of siboglinids in the Russian part of the Arctic (none yet found in the Chukchi Sea).

One location in the Laptev Sea coincides with the landfill where methane flares were registered [13].

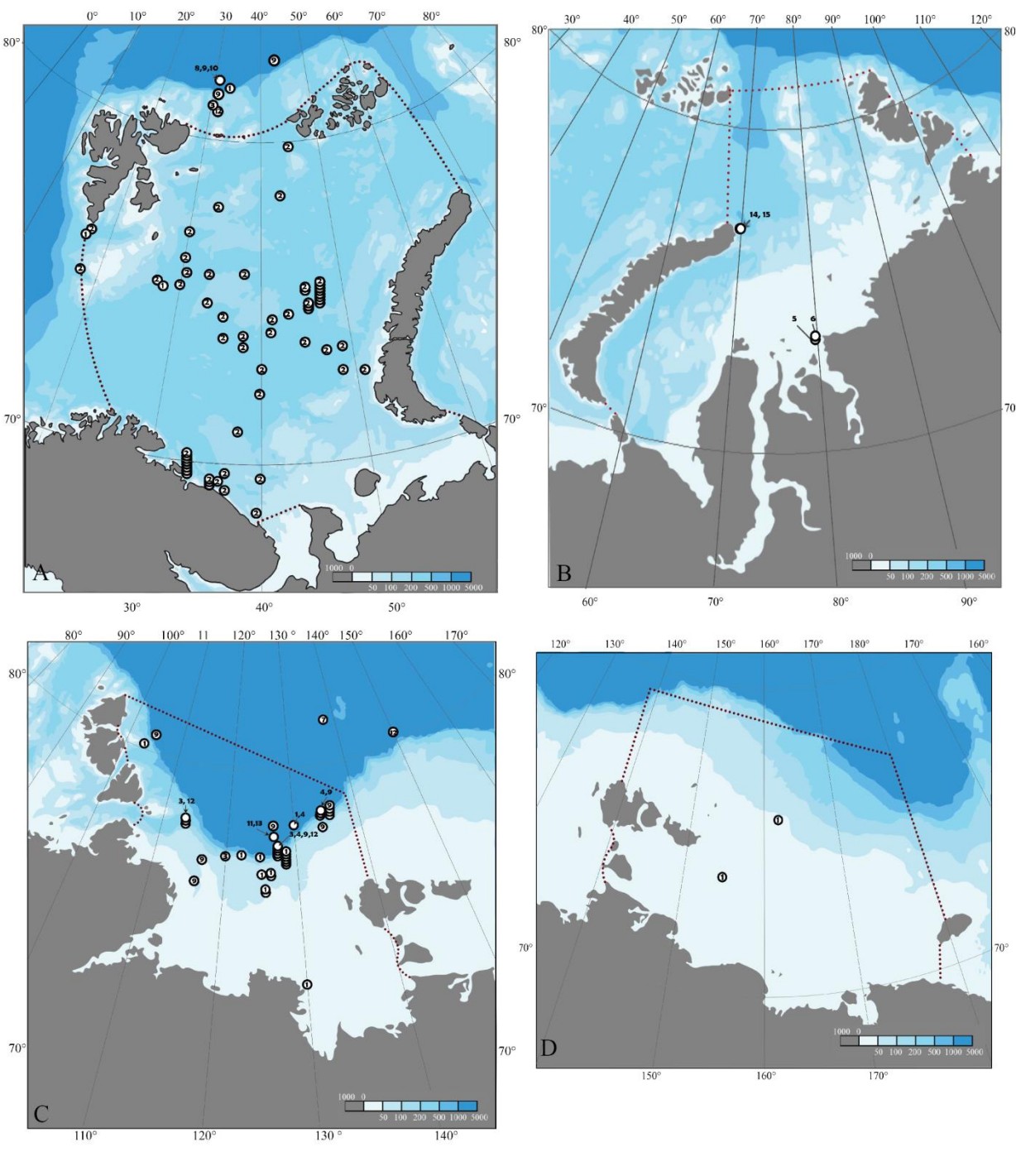

**Figure 1.** Distribution maps of siboglinid records in the Arctic seas of Russia. (**A**)—the Barents Sea, (**B**)—the Kara Sea, (**C**)—the Laptev Sea, (**D**)—the East Siberian Sea. The numbers indicate the records of the following siboglinid species: 1—*Oligobrachia haakonmosbiensis*, 2—*Nereilinum murmanicum*, 3—*Nereilinum squamosum*, 4—*Polarsternium rugellosum*, 5—*Crispabrachia yenisey*, 6—*Galathealinum karaense*, 7—*Polybrachia gorbunovi*, 8—*Siboglinum ekmani*, 9—*Siboglinum hyperboreum*, 10—*Siboglinum norvegicum*, 11—*Siboglinum* sp., 12—*Sclerolinum contortum*, 13—*Siboglinidae* sp. 1, 14—*Siboglinidae* sp. 2, 15—*Siboglinidae* sp. 3.

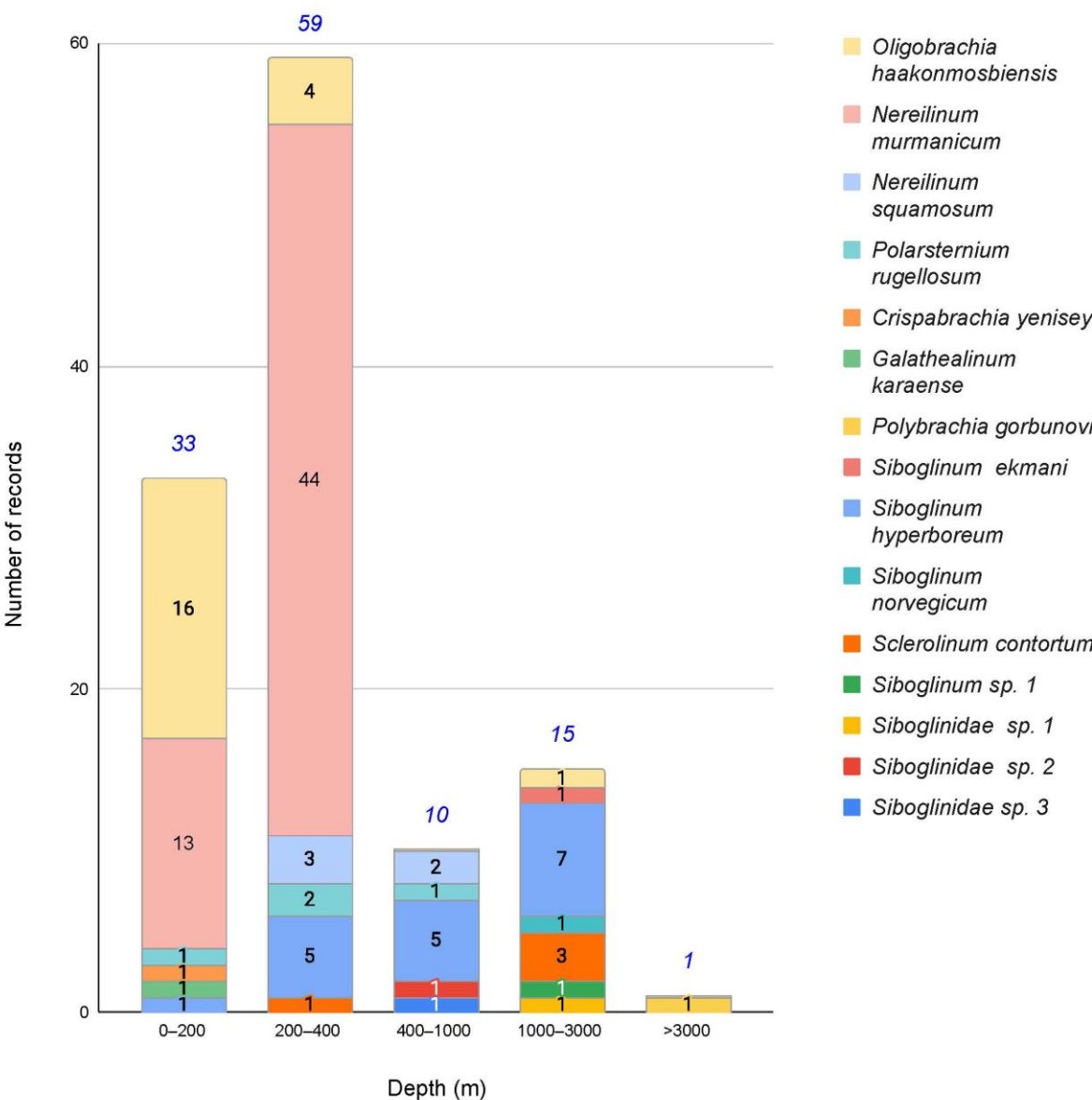

**Figure 2.** Distribution of siboglinid records by depth. The numbers indicate the number of records at a given depth range. The number of records of the corresponding species are indicated in black; the total number of siboglinid finds of the current depth range is indicated in blue italics.

Relatively recently, Sen and co-authors [50,51] demonstrated that *O. haakonmosbiensis* represents a complex of cryptic species that are morphologically indistinguishable. Molecular phylogenetic analysis enables the distinguishing of three separate species. The type species *O. haakonmosbiensis* is distributed in the Norwegian Sea at depths 721–1303 m [13,50]. The second species of *Oligobrachia* CPL-clade has a predominantly Arctic distribution from the Svalbard archipelago on the border of the Norwegian and Barents Seas to the East Siberian Sea at depths of 26–380 m [50,52,53]. The third species is *Oligobrachia* sp. Vestnesa: it currently has a point range and is known only from two pockmarks in the region of the Vestnes ridge to the northwest of the Svalbard archipelago at a depth of 1200 m, where the type species *O. haakonmosbiensis* is also found [51]. All the records considered by us in this publication, from the Barents to the East Siberian Sea, are attributable to one of three cryptic species, namely, *Oligobrachia* CPL-clade [50,51,53]. The only record of the *O. haakonmosbiensis* in the Arctic basin northeast of Svalbard cannot yet be reliably attributed to one of the three cryptic species; it represents the deepest record at a depth of 2166 m ([40], collection of ZIN RAS).

2. *Nereilinum murmanicum* Ivanov, 1961

Among the regions considered by us, representatives of the species *N. murmanicum* are known only in the Barents Sea, where it is widely distributed within 69°5′–78°03′ N and 16°25′–51°01′ E (Figure 1A). The northern boundary of the *N. murmanicum* habitat is the southeastern part of the Perseus upland. To the west, the area is limited by the Medvezhinsko-Nadezhdinskaya upland on the border of the Barents and Norwegian Seas. The southernmost point falls on the southwestern part of the Murmansk upland and almost approaches the coast of the Kola Peninsula. In the east, the range is limited by the Central depression and the southeastern coast of the Southern Island of the Novaya Zemlya archipelago. The depth range at which *N. murmanicum* was found in the Barents Sea is 75.2–375.3 m; many of the records stem from 200 to 300 m (Figure 2) [40,42,54,55]. A total of 57 stations with records of this species are known within the Barents Sea (Figure 1A). All the records in the Barents Sea, except one, are confined to soft soils (silt, silty sand, clay), and the vast majority are on various silty soils, with a small number on clay soils. One record stems from a rocky ground [42,56]. In addition to the Barents Sea, this species was also reported in the Norwegian Sea, where it inhabits much greater depths of 990–1300 m [57].

3. *Nereilinum squamosum* Smirnov, 1999

The species is known from only a few records in two Arctic regions. One record in the Arctic basin is in the Barents Sea sector at a depth of 603 m (Figure 1A). Four stations are known from the Laptev Sea (Figure 1C), distributed along the shelf lower boundary and on the continental slope at depths of 240–560 m. These localities coincide with the habitats of several other siboglinid species (*Polarsternium rugellosum* Smirnov, 1999, *Siboglinum hyperboreum* Ivanov, 1960, *Sclerolinum contortum* Smirnov, 2000).

4. *Polarsternium rugellosum* Smirnov, 1999

This species is known only from the northeastern part of the Laptev Sea within 126°–130° E and 77°–78° N (Figure 1C) at a depth range of 100–556 m (Figure 2). The records coincide with the habitats of other siboglinid species: *N. squamosum*, *S. hyperboreum*, *Sc. contortum*.

5. *Crispabrachia yenisey* Karaseva, Rimskaya-Korsakova, Ekimova, Gantsevich, Kokarev, Kremnyov, Simakov, Udalov, Vedenin & Malakhov, 2021

The species is known from a single record in the Kara Sea in the estuary of the Yenisei River between Sibiryakov Island and the western shore of the Taimyr Peninsula at a depth of 28 m (Figure 1B).

6. *Galathealinum karaense* Smirnov, Zaitseva & Vedenin, 2020

This species, as in the case of the above-described *C. yenisey*, was discovered at the mouth of the Yenisei River at a depth of 25 m [43], whereby the records of both species were made relatively close to each other (Figure 1B).

7. *Polybrachia gorbunovi* (Ivanov, 1949)

Initially, this species was described by A.V. Ivanov [58] under the name *Lamellisabella gorbunovi*. Later, it was transferred to the genus *Polybrachia* [59]. It is known from a single record in the Arctic Basin in the Sadko Trench, opposite the Novosibirsk Islands (Figure 1C). The species was discovered during trawling on muddy soils at a depth of 3700–3800 m [38,60]. This record is the deepest of all Arctic siboglinids (Figure 2).

8. *Siboglinum ekmani* Jägersten, 1956

In the Arctic, this species is known only from one site in the Arctic Basin to the northeast of the Svalbard archipelago outside the Barents Sea at a depth of 2090–2166 m (Figure 1A). This record coincides with the discovery site of two more species of the genus: *Siboglinum* Caullery, 1914 (*Siboglinum hyperboreum* Ivanov, 1960 and *Siboglinum norvegium* Ivanov, 1960).

The Arctic record of this species is located in both the northernmost and easternmost parts of its ranges. The range mostly occupies the northern part of the Atlantic Ocean. The records of *S. ekmani* are distributed along the continental slope of North America down to the Caribbean Sea in the west and from the Norwegian Sea to the coast of Portugal in the east [12,61–68]. The species is one of the most eurybate among the frenulate pogonophorans and inhabits a depth range of 90–4485 m [12,62,64,66–68].

9.     *Siboglinum hyperboreum* Ivanov, 1960

Within the Russian sector of the Arctic, representatives of this species were recorded in two areas in the Arctic Basin and the Laptev Sea. In the Arctic Basin, only two records are known from the northeast of Svalbard in the range of 30°–42° E to 80° N (Figure 1A). In the Laptev Sea, it is widely distributed from the west of the Severnaya Zemlya archipelago to the northeastern margin of the sea in the east (Figure 1C). In the Laptev Sea, the records stem from 55 to 2000 m. Most of the records in the Arctic occur at depths below 500 m (Figure 2). Some *S. hyperboreum* records in the Laptev Sea are located in the area with methane seeps [69].

Outside the Russian sector of the Arctic, the species has also been recorded in the Greenland Sea, with one record near the Greenland coast at 217 m and one in the Svalbard area at about 650 m [38,40,44,60,66]. Thus, *S. hyperboreum* has a purely Arctic range and eurybate distribution.

10.     *Siboglinum norvegicum* Ivanov, 1960

This species is known from only a few records; one of them is in the Arctic Basin northeast of Svalbard in the Barents Sea at a depth of 2090–2166 m and coincides with the habitat of the above-mentioned *S. ekmani* and *S. hyperboreum* (Figure 1A). In addition to the Arctic Basin, the species is known from the Norwegian Sea, where it was recorded near the Shetland Islands at 120 m and off the coast of Norway at 165 m [38,60].

11.     *Siboglinum* sp.

Representatives of *Siboglinum* sp. were recorded in the central part of the Laptev Sea at a depth of 1556–1560 m (Figure 1C). The species could not be determined due to the poor preservation of the material. At the same station, in addition to *Siboglinum* sp., a representative of Frenulata of the indeterminate genus *Siboglinidae* gen. sp. 1. was also found.

12.     *Sclerolinum contortum* Smirnov, 2000

*Sc. contortum* within the Arctic region is known from several records in the Arctic Basin and the Laptev Sea. Two records are known in the Arctic Basin, the first being in the area adjacent to the Barents Sea to the northeast of the Svalbard archipelago (Figure 1A) and the second being in the East Siberian Sea area north of the Novosibirsk Islands (Figure 1C,D). The Arctic Basin records are confined to depths of 2000–2100 m (2). In the Laptev Sea, the species was found twice in its eastern and central parts at 310 and 1080 m. In three out of four cases, the presence of *Sc. contortum* coincides with the records of *N. squamosum* (Figure 1A,C).

*Sc. contortum* has a wide distribution outside the Arctic. It was found in the Norwegian Sea in the area of the Loki's castle hydrothermal field within the mid-Arctic ridge at its maximum depths of 2340–2740 m [13,70,71]. It also occurs in the Gulf of Mexico [15,71]. The southernmost area of discovery is in the Atlantic sector of the Southern Ocean near the South Shetland and South Sandwich Islands.

13.     *Siboglinidae* gen. sp. 1

These representatives of frenulate pogonophorans of uncertain taxonomic rank were found in the central part of the Laptev Sea at a depth of 1556–1560 m together with the undescribed species *Siboglinum* sp. mentioned above (Figure 1C). According to molecular phylogenetic analysis, *Siboglinidae* gen. sp. 1 is a sister genus to *Crispabrachia* [45].

14. *Siboglinidae* gen. sp. 2

A specimen with a single tentacle in a smooth tube was found in the Kara Sea near the northern tip of the Novaya Zemlya archipelago in the St. Anna Trench at a depth of 535.5 m (Figure 1B) [47]. The material is not described due to poor (incomplete) preservation.

15. *Siboglinidae* gen. sp. 3

A single specimen with two tentacles in a ringed tube was found in the Kara Sea in the St. Anna Trench together with *Siboglinidae* gen. sp. 2 indicated above (Figure 1B) [47]. The material is not described due to poor preservation.

## 4. Discussion

The total number of siboglinid records in the Arctic seas of Russia and the adjacent sector of the central basin of the Arctic Ocean is 120. The largest number of records (59) falls on the Barents Sea (49.2%), followed by the Laptev Sea (45; 37.5%) (Figure 3). The Kara Sea accounts for 3.3% of the records, and the East Siberian Sea accounts for 1.7% (Figure 3).

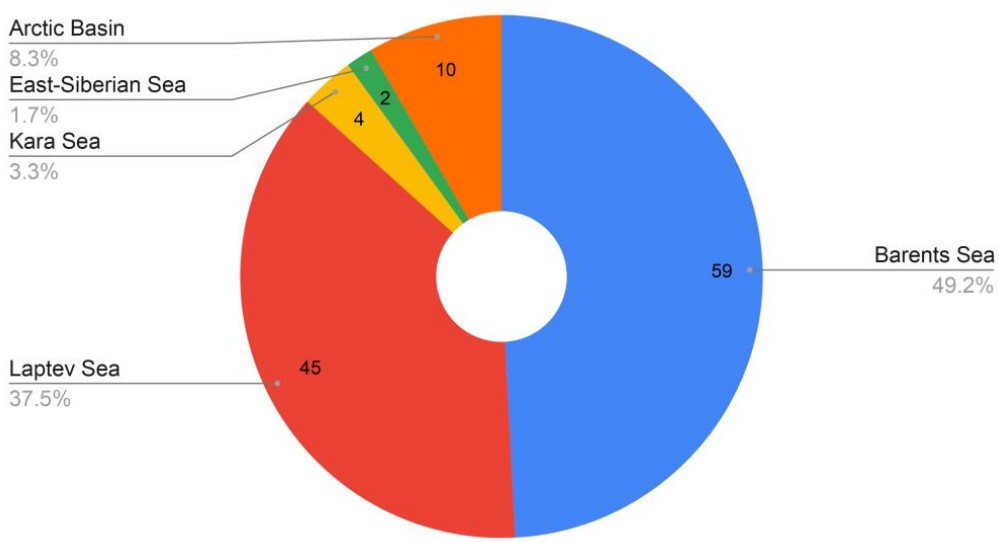

**Figure 3.** Distribution of siboglinid records by seas.

No siboglinids have yet been reported from the Chukchi Sea. Ten records are known in the areas adjacent to the central Arctic Basin (8.3%). Based on current knowledge, the distribution of the number of records most likely reflects the study effort in the various Arctic parts. Nonetheless, one fact is clear. The largest number of siboglinid records has been recorded in the Barents Sea (probably reflecting the best knowledge of this region), while the largest number of species was found in the Laptev Sea and the Arctic Basin (Figure 4).

Siboglinids are mainly deep-sea organisms whose representatives mostly inhabit bathyal depths [38,39,72]. In the Russian Arctic, most of the siboglinid records stem from the shelf (Figure 5), with 28% of all records at depths shallower than 200 m. The Barents Sea is characterized by a deep shelf, extending to depths of 300–400 m, reflecting a great glaciation setting in, followed by a sinking of the land in Quaternary time [73–76]. In this regard, most of the siboglinid records in the Barents Sea at depths down to 400 m are attributable to shelf habitats. In the Russian sector of the Arctic, 78% of the records are at depths down to 400 m (Figure 6). Taking into account our poor knowledge of the deep-water part of the Arctic Ocean, we can expect a higher number of siboglinid records at bathyal depths.

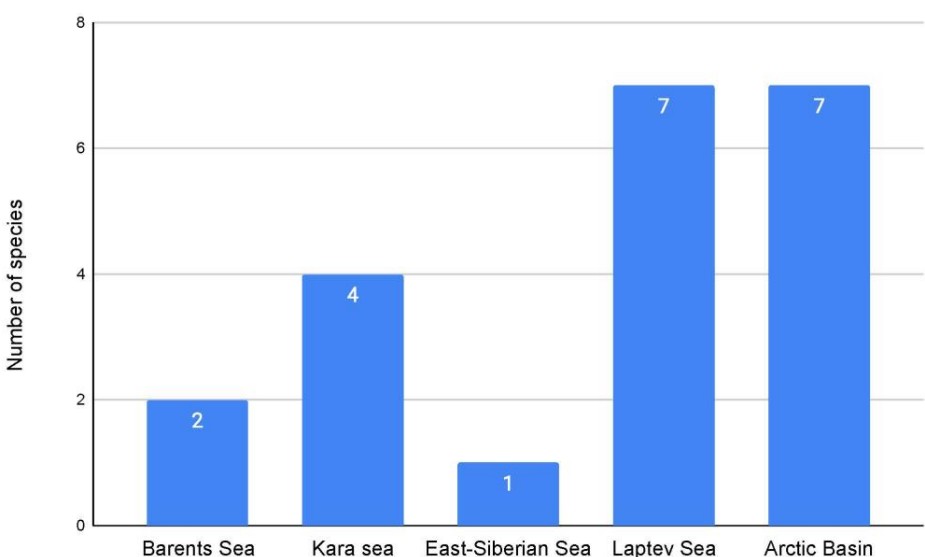

**Figure 4.** Distribution of siboglinid species by seas.

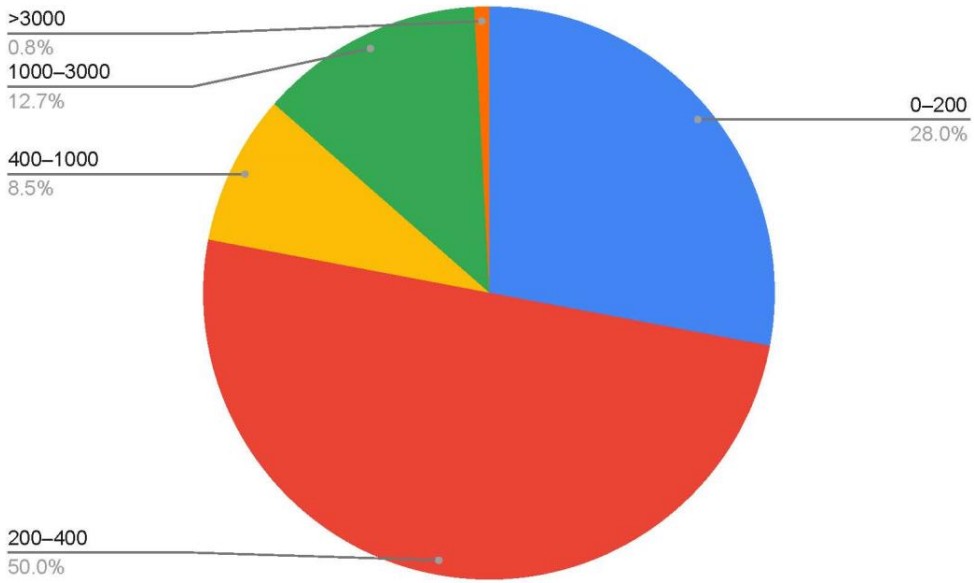

**Figure 5.** Percentage distribution of siboglinid records by depth.

Many of the siboglinid records in the Russian Arctic sector are associated with areas featuring high hydrocarbon concentrations. An earlier analysis of the distribution of *N. murmanicum* showed that 82% of the total records of this species in the Barents Sea correspond to areas known as highly promising for oil and gas production [56,77]. *N. murmanicum* was found near the largest gas fields of the Barents Sea, namely, Shtokman, Ledovskoye, and Ledovoye [56].

The main hydrocarbon reserves in the world's oceans are concentrated in the form of methane gas hydrates [35,78–97]. Many siboglinid records in the Barents Sea and the adjoining Arctic Basin are confined to the areas of such gas hydrates (Figure 6A). The records of *N. murmanicum* are confined to gas hydrate deposits in the central part of the Barents Sea. The gas hydrate areas in the western part of the Barents Sea are associated with the records of both *N. murmanicum* and *O. haakonmosbiensis* (Figure 6A). The marginal gas hydrate deposit sites in the north Barents Sea and the adjoining Arctic basin, where methane emissions can occur, are associated with the records of *O. haakonmosbiensis*, *N. murmanicum*, *S. ekmani*, *S. hyperboreum*, and *S. norvegicum* (Figure 6A).

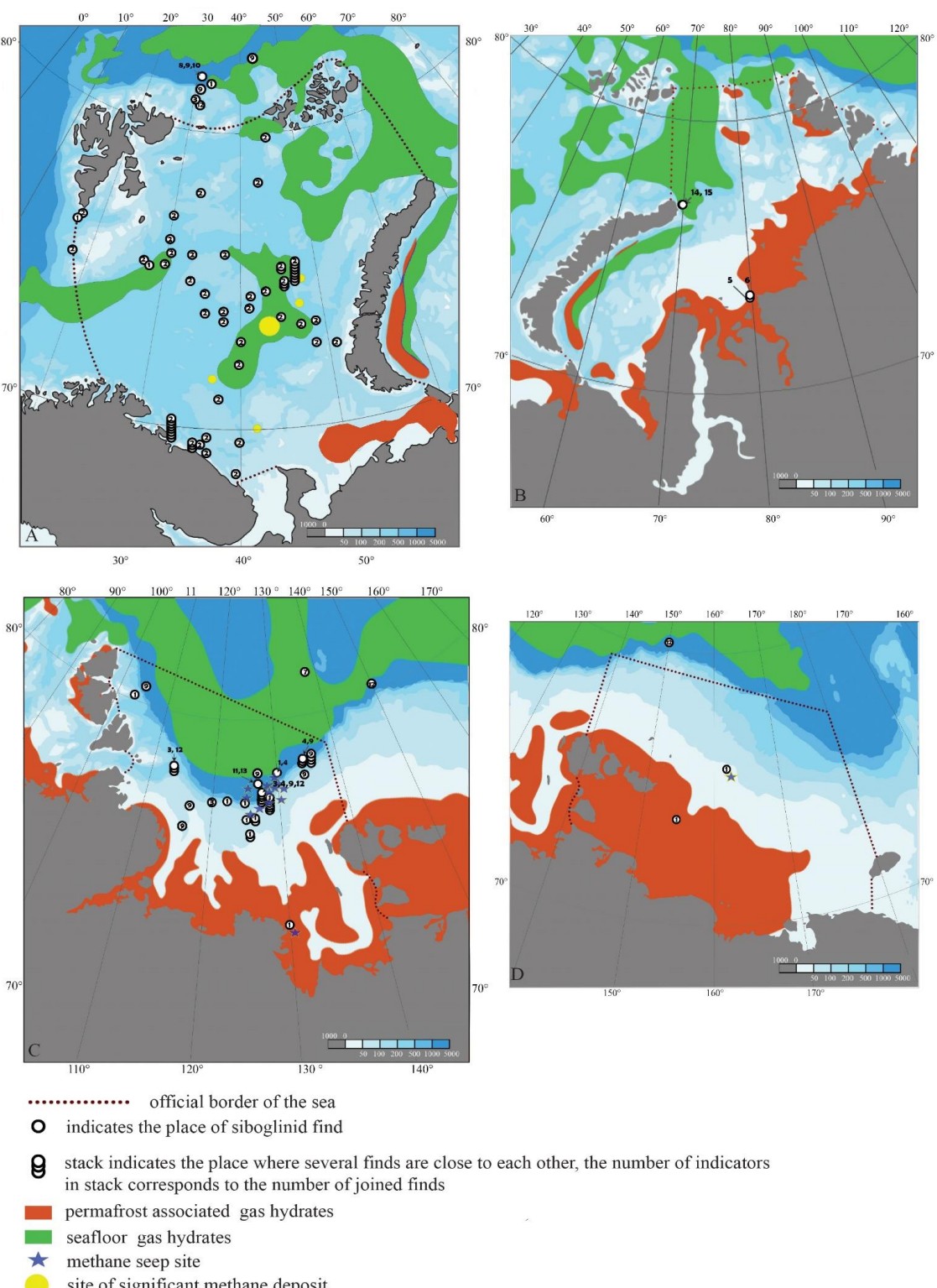

**Figure 6.** Distribution map of siboglinid records in the Arctic seas of Russia with different methane occurrences (deposits, gas hydrates (by Solovyov et al., 1987 [98]), and methane seep sites (by Shakhova et al., 2015 [69])). (**A**)—the Barents Sea, (**B**)—the Kara Sea, (**C**)—the Laptev Sea, (**D**)—the East-Siberian Sea. The numbers indicate the records of the following siboglinid species: 1—*Oligobrachia haakonmosbiensis*, 2—*Nereilinum murmanicum*, 3—*Nereilinum squamosum*, 4—*Polarsternium rugellosum*, 5—*Crispabrachia yenisey*, 6—*Galathealinum karaense*, 7—*Polybrachia gorbunovi*, 8—*Siboglinum ekmani*, 9—*Siboglinum hyperboreum*, 10—*Siboglinum norvegicum*, 11—*Siboglinum* sp., 12—*Sclerolinum contortum*, 13—*Siboglinidae* sp. 1, 14—*Siboglinidae* sp. 2, 15—*Siboglinidae* sp. 3.

Gas hydrate deposits in the Russian Arctic are divided into bottom gas hydrates, which occupy deep-water areas of the Arctic Ocean, and gas hydrate deposits in the permafrost, which are located at shallow depths relatively close to the Russian Arctic coast [35,81,90,98–104]. Nonetheless, the bottom and permafrost gas hydrate areas in most of the Arctic seas of Russia are separated by extensive zones where gas hydrates are absent. This reflects either a lack of methane or a lack of thermobaric conditions necessary for the formation of clathrates.

The Kara Sea surpasses all other seas of the Russian Arctic in terms of hydrocarbon resources [31,105,106]. The Kara Sea siboglinid fauna remains very poorly studied. Frenulate pogonophorans have so far been found in only two areas here (Figure 6B). One of them is in the Yenisei Bay between Sibiryakov Island and the western shore of the Taimyr Peninsula [41,43,45]. In the former, both records stem from record-shallow depths for siboglinids. *Crispabrachia yenisey* was recorded at a depth of 28 m [41,45], while *Galathealinum karaense* was found at 25 m [43]. Siboglinids are stenohaline organisms and, as a rule, are not found in less salty waters [38]. The Yenisei Bay is characterized by strong a vertical stratification of salinity [107–109]. The surface average long-term value between Sibiryakov Island and the western coast of the Taimyr Peninsula around the records of *C. yenisey* and *G. karaense* is less than 5‰, whereas the bottom salinity here ranges from 30 to 32.5 ‰ [107–109]. The discovery site of *C. yenisey* and *G. karaense* is in an area where the methane concentration in the surface water layer reaches 130 nM, a peak value for the southern part of the Kara Sea [110]. The high concentrations of methane here reflect the degradation of permafrost gas hydrates under the influence of river runoff [100,110,111]. This process is intensive against the background of the Arctic general warming in the estuaries of Ob, Yenisei, Lena, and other large rivers of the Russian Arctic. The result is not only high concentrations of methane in the water but also the release of this greenhouse gas into the atmosphere [102,103,112–118].

A species closely related to *G. karaense*, namely, *Galathealinum arcticum* Southward, 1962, was found in the Canadian Arctic in the estuary area of the Mackenzie River at a depth of 36 m [119]. The estuarine areas of that river are also characterized by very strong salinity stratification: the values in the surface layer range from 1 to 10‰, but at 20 m, they exceed 31‰ in all seasons [120]. The Mackenzie River Delta and adjacent areas of the Beaufort Sea shelf are characterized by large gas hydrate deposits in the permafrost, which dissociate under the influence of river runoff under the conditions of Arctic warming and generate powerful methane flows [121–130].

The area in which *Siboglinidae* gen. sp. 2 and *Siboglinidae* gen. sp. 3 were found (below 500 m depth, salinity exceeding 34‰) [47] corresponds to the southernmost section of the bottom gas hydrate distribution in the St. Anna Trench [98,99,104]. Warm and salty Atlantic water flows through the Fram Strait into the central depression of the Arctic Ocean and, further along the trench, enters the Kara Sea [46,131–134]. Existing models [35,135,136] demonstrate the potential dissociation of bottom gas hydrates in the St. Anna Trench, and the resulting methane stream serves as a source that ensures the survival of siboglinids.

The Laptev Sea also harbors some of the richest deposits of hydrocarbons [69,103,112,113,137–139]. Its southern part is characterized by gas hydrate deposits in the permafrost, whereas the deep-water northern part features extensive deposits of bottom gas hydrates [99]. Numerous methane seeps have been recorded on the outer part of the shelf in the northeast Laptev Sea [69,103,112,113]. To date, seven species of siboglinids have been reported in the Laptev Sea (including undetermined records), which is more than that in any other Russian Arctic sea (Figure 4). The distribution of siboglinids in the Laptev Sea is clearly correlated with areas of methane seeps of various geneses. The most widespread species in the Russian Arctic, *O. haakonmosbiensis,* was found in the Lena River estuary (Figure 6C), where the methane concentration of the water exceeds 1000 nm [117]. As in the case of the Yenisei and Mackenzie River mouths, high methane concentrations here reflect the degradation of permafrost under the influence of river runoff [69,103,110,117,138,140]. Siboglinid records in the northern part of the sea are

associated with marginal areas of bottom gas hydrate deposits (Figure 6C). The discovery of *P.gorbunovi* in a section of the Arctic Basin adjacent to the Laptev Sea is associated with deep-sea gas hydrates (Figure 6C). The most numerous siboglinid records are confined to the field of methane seeps on the outer part of the Laptev Sea shelf (Figure 6C): all seven species recorded to date in the Laptev Sea were found in this area [139,141,142].

The East Siberian Sea is also among the most hydrocarbon-rich seas of the Russian Arctic. Methane seeps from underwater deposits cause high methane concentrations in the water column and strong emissions into the atmosphere [102,103,110,112,113,117,143,144]. Unfortunately, the siboglinid fauna of the East Siberian Sea remains poorly studied. Only two records of *O. haakonmosbiensis* are known here: on the shelf sections southeast and southwest of New Siberia Island (Figure 1D). One of those records was made at a depth of 26 m and is in the gas hydrate zone associated with coastal permafrost strata (Figure 6D). The methane emissions here are with the dissociation of gas hydrates of permafrost [69,102,103,110,115,117,137,138,145]. Another record at a depth of 48 m in the central part of the sea (Figure 6D) coincides with the landfill where methane flares have been registered [144]. In the deep-water part of the Arctic Basin adjacent to the East Siberian Sea, the presence of *Sclerolinum contortum* coincides with the marginal area of bottom gas hydrates.

The presented material shows a correlation of siboglinids records with various types of methane manifestations. As noticed above, in the Laptev Sea and in the Yenisei Bay, there are certain data on increased methane concentrations in places coinciding with the recordings of siboglinids [110,117]. In the East Siberian Sea and the Laptev Sea, some records coincide with methane flares [139,144]. For a general assessment of the possible effect of gas hydrates on the distribution of siboglinids across all seas of the Arctic seas of Russia, we present Solovyov's data on the areas of gas hydrate content [96]. It should be noticed that methane can also come from many other sources—for example, from organic residues (such as whale falls and flooded wood). The presence of communities on submerged wood is very likely, at least in the area of *P. gorbunovi* records, since mollusks associated with submerged wood were collected at the same station [146,147].

## 5. Conclusions

Rich methane deposits have been identified in the Arctic seas of Russia and the adjacent part of the Arctic basin. This suggests the presence of a rich siboglinid fauna: gutless worms that live in symbiosis with chemoautotrophic bacteria that oxidize methane or hydrogen sulfide. Hydrogen sulfide is formed in areas of hydrocarbon seeps due to the anaerobic oxidation of methane in the sediment. Overall, the fauna of the Arctic seas and the Arctic Basin is not well studied. Nevertheless, siboglinids have been recorded in the basin and in all Arctic seas of Russia, except for the Chukchi Sea. The range of siboglinids is associated with hydrocarbon manifestations of various geneses. In the Barents Sea, most of the siboglinid records are associated with large gas and oil fields such as Shtokman, Ledovskoye, and Ledovoye. Elsewhere, they are associated with the marginal areas of bottom gas hydrates in the Arctic Basin, where hydrate dissociation generates methane flows. In the Laptev Sea and the East Siberian Sea, most of the siboglnid records are associated with areas of methane flares. The siboglinid records in the estuaries of large Siberian rivers, such as the Yenisei and Lena, are of particular interest. These records are associated with the emission of methane from permafrost rocks under the influence of river runoff.

**Author Contributions:** Conceptualization, N.P.K. and V.V.M.; Methodology, R.V.S. and A.A.U.; Validation, V.O.M. and M.M.G.; Formal analysis, N.P.K. and V.V.M.; Investigation, N.P.K., N.N.R.-K. and V.V.M.; Resources, R.V.S. and A.A.U.; Data curation, R.V.S. and A.A.U.; Writing—original draft preparation, N.P.K. and V.V.M.; Writing—review and editing, N.P.K., M.M.G., and V.V.M.; Visualization, N.P.K.; Supervision, V.V.M.; Project administration, V.V.M.; Funding acquisition, V.V.M. All authors have read and agreed to the published version of the manuscript.

**Funding:** This research was funded by the Russian Science Foundation, grant number 18-14-00141-P.

**Institutional Review Board Statement:** Not applicable.

**Data Availability Statement:** Not applicable.

**Acknowledgments:** The authors are grateful to the colleagues from the P.P. Shirshov Institute of Oceanology Russian Academy of Sciences and the Zoological Institute, Russian Academy of Sciences for the availability of the collection data. We would like to thank Andrey Vedenin, Valentin Kokarev, and Miloslav Simakov for the work on the research cruises and the help in accessing the material. The authors thank the anonymous reviewers for their valuable comments and suggestions which allowed for the considerable improvement of the manuscript.

**Conflicts of Interest:** The authors declare no conflict of interest.

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
