# Peer review of "Distribution of Gutless Siboglinid Worms (Annelida, Siboglinidae) in Russian Arctic Seas in Relation to Gas Potential"

_diversity, doi:10.3390/d14121061_

Round 1

Reviewer 1 Report

In my opinion, the manuscript is an original and interesting study and can be published in the journal Diversity.

However, I would make a few comments to the authors, the correction of which, in my opinion, can improve the manuscript.

1. The attempt to link the Siboglinid findings to areas of high methane content according to the map from Solovyov's work does not look very convincing. This map is very generalized and according to it, places with high methane content occupy up to 1/3 of the area of the Arctic seas. This slightly distorts the reality: indeed, methane of different origins CAN occur in these areas, but it is absolutely not necessary that it is present there in any significant concentrations. It is best to rely on real data from measurements of methane concentrations in worm habitats. If there is no such data, then it should be indicated that the worms themselves can be indicators of the presence of chemosynthetic communities in this place, regardless of the Solvoyev map.

2. Methane yields of various origins are not the only type of chemosynthetic communities in the Arctic. At least whale bones and sunken wood can be found here. Both types of substrates release a large amount of substances during decomposition, including CH4, NH3, SH, etc. These gases are used by bacteria and allow chemosynthetic communities to exist. The presence of communities on submerged wood is very likely, at least at the Polybrachia gorbunovi locality, since mollusks associated with submerged wood were collected at the same station. Please see these papers:

Krol E.N., Nekhaev I.O. 2020. Redescription of Leptogyra bujnitzkii (Gorbunov, 1946) comb. nov., the first representative of the gastropod subclass Neomphaliones from the high Arctic. Zootaxa, 4759 (3): 446-450. https://doi.org/10.11646/zootaxa.4759.3.13

Nekhaev I.O. 2022. Skenea profunda (Vetigastropoda: Skeneidae) in the central Arctic. Ruthenica, 32(2): 105-109 https://doi.org/10.35885/ruthenica.2022.32(3).2 

Author Response

Dear Reviewer,

We are grateful for your work on reviewing, as well as for valuable comments that help improve our manuscript.

In the corrected version, we tried to take into account all your comments as much as possible and added additional information.

Moreover to corrections below you can find detailed answer to your comments

  1. The attempt to link the Siboglinid findings to areas of high methane content according to the map from Solovyov's work does not look very convincing. This map is very generalized and according to it, places with high methane content occupy up to 1/3 of the area of the Arctic seas. This slightly distorts the reality: indeed, methane of different origins CAN occur in these areas, but it is absolutely not necessary that it is present there in any significant concentrations. It is best to rely on real data from measurements of methane concentrations in worm habitats. If there is no such data, then it should be indicated that the worms themselves can be indicators of the presence of chemosynthetic communities in this place, regardless of the Solvoyev map”

We make a few corrections in the text according to your comments.

We absolutely agree that it is much better to rely on real data from measurements of methane concentrations and we provide references to the works where such data are available. In our case it is impossible to provide such data for each of the siboglinid finds.  We took the map of Soloviev et al. 1987 as one of the general and most easy to reproduce as otherwise maps will be too difficult to read. On other hand we can see a lot of other works, where we find same or mostly same areas of hydrates as that one showed in Soloviev et al., 1987

(for example:      C. Ruppel Permafrost-Associated Gas Hydrate: Is It Really Approximately 1 % of the Global System? Journal of Chemical & Engineering Data 2015 60 (2), 429-436 DOI: 10.1021/je500770m;  Giustiniani M, Tinivella U, Jakobsson M, Rebesco M: Arctic ocean gas hydrate stability in a changing climate. Journal of Geological Research 2013(783969):1-10. http://dx.doi.org/10.1155/2013/783969;         Andreassen, Karin, Alun Hubbard, M. Winsborrow, Henry Patton, Sunil Vadakkepuliyambatta, Andreia Plaza-Faverola, Eythor Gudlaugsson et al. "Massive blow-out craters formed by hydrate-controlled methane expulsion from the Arctic seafloor." Science 356, no. 6341 (2017): 948-953. DOI: 10.1126/science.aal4500;      and many others which we cite in our paper and which data we took in mind when providing statements of this work).

We think that it is quite obvious that there could be chemosynthetic communities in places of Siboglinid finds and this statement doesn’t need to be postulated.

  1. “Methane yields of various origins are not the only type of chemosynthetic communities in the Arctic. At least whale bones and sunken wood can be found here. Both types of substrates release a large amount of substances during decomposition, including CH4, NH3, SH, etc. These gases are used by bacteria and allow chemosynthetic communities to exist. The presence of communities on submerged wood is very likely, at least at the Polybrachia gorbunovi locality, since mollusks associated with submerged wood were collected at the same station. Please see these papers:
  •         Krol E.N., Nekhaev I.O. 2020. Redescription of Leptogyra bujnitzkii (Gorbunov, 1946) comb. nov., the first representative of the gastropod subclass Neomphaliones from the high Arctic. Zootaxa, 4759 (3): 446-450. https://doi.org/10.11646/zootaxa.4759.3.13
  •         Nekhaev I.O. 2022. Skenea profunda (Vetigastropoda: Skeneidae) in the central Arctic. Ruthenica, 32(2): 105-109 https://doi.org/10.35885/ruthenica.2022.32(3).2 “

 Among Siboglinids there are two genera: Osedax and Sclerolinum which are common on whale or other bones and wood, while Frenulates, which are the most finds in the Arctic, often inhabit soft silty sediments. In our work we wish to underline that methane of different sources including “geological” (i.e. gas hydrates of different source) also influenced the distribution of chemosynthetic organisms such as Siboglinids. 

According to your note we provided changes in the last paragraph of Discussion.

Reviewer 2 Report

This manuscript reports the distribution of siboglinid annelid species in the Russian arctic seas and adjacent areas in the Arctic basin. This manuscript provides a detailed catalog of several species present in these seas. Here are some of my comments. 

1) The rationale behind this work remained unclear to me. It would be great, if the authors can state what they wish to accomplish from this catalog of distribution of these species in the Arctic seas. In what ways would this catalog be useful?

2) Next, it was also unclear to me which species have been reported and described before and which are new findings by the authors in this manuscript. Please make a table or a summary figure of what species you found and which ones were previously reported. 

3) Are there any information regarding phylogenetic relationships between these species? Please comment on phylogenetic relationships of these species and their evolutionary history. How does their geographic distribution correlate with their genetic variation? Are there any evidence of that? Did the authors look into that?

With these few changes I recommend this manuscript for publication

Author Response

Dear Reviewer,

We are grateful for your work on reviewing, as well as for valuable comments that help improve our manuscript.

In the corrected version, we tried to take into account all your comments as much as possible and added additional information.

Moreover  to corrections below you can find detailed answer to your comments

1)” The rationale behind this work remained unclear to me. It would be great, if the authors can state what they wish to accomplish from this catalog of distribution of these species in the Arctic seas. In what ways would this catalog be useful”

We add our statements into the introduction, please check the last paragraph.

2) Next, it was also unclear to me which species have been reported and described before and which are new findings by the authors in this manuscript. Please make a table or a summary figure of what species you found and which ones were previously reported. 

We provide corrections into the “Data sources” part to make this clear.

3) Are there any information regarding phylogenetic relationships between these species? Please comment on phylogenetic relationships of these species and their evolutionary history. How does their geographic distribution correlate with their genetic variation? Are there any evidence of that? Did the authors look into that?

Currently our state of knowledge on frenulate Siboglinids, which are most of the Arctic Siboglinid finds, doesn't allow us to make extended comments on this subject. Oligobrachia haakonmosbiensis – the only species, about which some comments on phylogenetic relationships and genetic variation could be done. This information is included in the manuscript.

Round 2

Reviewer 1 Report

No comments or suggestions

Author Response

Dear Reviewer

We are grateful for your work on reviewing. We make some final corrections into the text and figures. Also we've asked for advice of an English-speaking person to improve text and hope that it has become more clear.

You can find corrected version into attachment

Sincerely yours, Nadezda
